# Catheter Ablation for Atrial Fibrillation in Patients with Heart Failure with Preserved Ejection Fraction: A Systematic Review and Meta-Analysis

**DOI:** 10.3390/jcm11020288

**Published:** 2022-01-06

**Authors:** Emmanuel Androulakis, Catrin Sohrabi, Alexandros Briasoulis, Constantinos Bakogiannis, Bunny Saberwal, Gerasimos Siasos, Dimitris Tousoulis, Syed Ahsan, Nikolaos Papageorgiou

**Affiliations:** 1Cardiovascular Imaging Department, Royal Brompton & Harefield Hospital NHS Foundation Trust, London SW3 6NP, UK; 2Cardiology Department, St George’s University of London, London SW17 0RE, UK; 3Electrophysiology Department, Barts Heart Centre, St. Bartholomew’s Hospital, London EC1A 7BE, UK; csohrabi42@gmail.com (C.S.); b.saberwal@nhs.net (B.S.); drnpapageorgiou@yahoo.com (N.P.); 4Alexandra Hospital, 80 Vasilissis Sophias Avenue, 11528 Athens, Greece; alexbriasoulis@gmail.com; 53rd Cardiology Department, AUTH, Ippokrateion Hospital, 55642 Thessaloniki, Greece; bakogianniscon@gmail.com; 63rd Cardiology Department, Sotiria Hospital, Athens University Medical School, 11527 Athens, Greece; ger_sias@hotmail.com; 71st Cardiology Department, Hippokration Hospital, Athens University Medical School, 11527 Athens, Greece; drtousoulis@hotmail.com; 8Institute of Cardiovascular Science, University College London, London WC1E 6BT, UK; syedyahsan@gmail.com

**Keywords:** heart failure with preserved ejection fraction, catheter ablation, pharmacological therapy, outcomes

## Abstract

Background: Catheter ablation (CA) for atrial fibrillation (AF) has been proposed as a means of improving outcomes among patients with heart failure and reduced ejection fraction (HFrEF) who are otherwise receiving appropriate treatment. Unlike HFrEF, treatment options are more limited in patients with preserved ejection fraction (HFpEF) and the data pertaining to the management of AF in these patients are controversial. The aim of this systematic review and meta-analysis was to investigate the effects of CA on outcomes of patients with AF and HFpEF, such as functional status, post-procedural complications, hospitalization, morbidity and mortality, based on data from observational studies. Methods: We systematically searched the electronic databases MEDLINE, PUBMED, EMBASE and the Cochrane Library for Central Register of Clinical Trials until May 2020. Results: Overall, the pooling of our data showed that sinus rhythm was achieved long-term in 58.0% (95% CI 0.44–0.71). Long-term AF recurrence was noticed in 22.3% of patients. Admission for HF occurred in 6.2% (95% CI 0.04–0.09) whilst all-cause mortality was identified in 6.3% (95% CI 0.02–0.13). Conclusion: This meta-analysis is the first to focus on determining the benefits of a rhythm control strategy for patients with AF and HFpEF using CA, suggesting it may be worthwhile to investigate the effects of a CA rhythm control strategy as the default treatment of AF in HFpEF patients in randomized trials.

## 1. Introduction

The co-existence of heart failure (HF) with atrial fibrillation (AF) confers a particularly poor prognosis, and the combination is frequently encountered given the overlap of predisposing risk factors, including older age, hypertension, metabolic syndrome, and diastolic dysfunction [1]. The causality between these entities is an area of ongoing research, but it follows that those patients with HF are likely to benefit from being in sinus rhythm versus AF, as evidenced by improvements in functional status and potential mortality benefits [2]. This relationship appears to hold for the various HF subtypes, including both HF with preserved (HFpEF) and HF with reduced (HFrEF) ejection fraction cohorts [1].

HFpEF is defined as the presence of typical signs and symptoms of congestion in the presence of a left ventricular ejection fraction (LVEF) >50% [3]. It has grown to become the most prevalent form of HF, rising by 10% each decade with respect to its HFrEF counterpart [4]. Whilst the diagnosis can be challenging to make in certain circumstances, biomarker assessment of N-terminal pro-brain natriuretic peptide (NT-proBNP) is considered to have increased diagnostic accuracy in detecting HFpEF in AF patients [5]. There are many similarities in the pathophysiological mechanisms between these two conditions, with structural remodeling and fibrosis seen both in atrial and ventricular myocardium leading to changes in electrical conduction and diastolic dysfunction, respectively [6]. The increased afterload caused by aldosterone leads to further myocardial fibrosis, and its levels have been shown to reduce after successful cardioversion of AF [7]. Notably, there are various factors that have been reported to predict mortality and arrhythmia recurrence outcomes in patients with AF, including functional mitral and tricuspid regurgitation [8], reduced left atrial ejection force [9], and QTc duration [10].

Unlike HFrEF, treatment options are more limited with diuretic therapy and mineralocorticoid antagonists representing the mainstay [11]. A concomitant diagnosis of AF appears to worsen outcomes more than either condition alone, and the data pertaining to the management of AF in patients with HFpEF is somewhat controversial [12]. Although catheter ablation (CA) success rates are limited in that approximately 60% or more for paroxysmal atrial fibrillation and 30% or less for persistent atrial fibrillation after a single procedure, success rates increase for multiple procedures.

In the absence of direct head-to-head randomized controlled trials of CA versus non-invasive treatment with rate control and/or antiarrhythmics, this meta-analysis focuses on assessing the impact of catheter ablation (CA) on outcomes of patients with AF and HFpEF, including functional status (such as New York Heart Association (NYHA) class), post-procedural complications, hospitalization, and morbidity and mortality data, based on data from observational studies.

## 2. Materials and Methods

### 2.1. Search Strategy

We systematically searched the electronic databases MEDLINE, PUBMED, EMBASE and the Cochrane Library for Central Register of Clinical Trials, using the MESH terms, ‘atrial fibrillation’ AND ‘ablation’ OR ‘catheter ablation’ AND ‘heart failure’ OR ‘heart failure with preserved ejection fraction’ OR ‘HFpEF’. We limited our search to studies in human subjects and English language in peer-reviewed journals published until May 2020. Additionally, a manual search of all relevant references from the screened articles and reviews was performed for additional clinical studies. The population, intervention, comparison, and outcome approach was used [13]. The population of interest included patients with HFpEF, and the intervention was CA of AF, or rhythm control with anti-arrhythmic drugs (AAD) [14,15,16,17]. In the absence of a control group, a non-controlled observational analysis was also performed. The primary outcome measure was AF recurrence post ablation. Procedural success was defined as freedom of AF (at the end of follow-up after a single ablation procedure). Other outcomes included sinus rhythm post ablation with or without AAD, change of symptoms, HF admission, all-cause admission, and mortality. Assessed procedural complications were procedural death, stroke, cardiac tamponade, acute myocardial infarction, major vascular complications, and major bleeding, assessed on a study-by-study basis.

In order to be included, studies were required to provide a minimum set of information regarding the sample of HFpEF patients undergoing CA of AF, namely age, gender, as well as information on the HFpEF diagnosis criteria, and baseline medication. Observational non-controlled case series studies required a minimum of five patients to be considered eligible. Review articles, editorials and case reports were not considered eligible for the purpose of this review. Reference lists of all accessed full-text articles were further searched for sources of potentially relevant information. Authors of full-text papers were also contacted by email to retrieve additional information if required.

### 2.2. Inclusion Criteria

We included prospective studies and retrospective cohorts published as original articles in peer-reviewed scientific journals in English. We did not restrict eligibility according to renal function.

### 2.3. Exclusion Criteria

We excluded those trials that did not report any of the following variables or outcomes: number of events in both the intervention and reference groups, length of study, description of the main relevant features of the study population, including gender, age, description of the procedure and concomitant therapy.

### 2.4. Data Extraction and Quality

Data extraction and presentation followed recommendations as established by the PRISMA group. Where available, the following data were extracted to allow characterization of each patient sample, including study design, study population characteristics (age, gender, body mass index-BMI, and co-morbidities), AAD, AF type, LVEF, NYHA class, NT-pro-BNP, eGFR/creatinine, duration of AF prior to intervention, and follow-up duration. Specific data regarding ablation type (radiofrequency and cryoballoon), left atrium (LA) dimension, LA volume, LV mass, interventricular septum end diastolic dimension (IVSd), LV end diastolic dimension (LVEDd), mitral inflow velocities (E/A), E/E’ and procedure time were also collated.

The data were independently extracted by two authors using a standardized protocol and reporting form. Two independent reviewers (E.A. and N.P.) screened all abstracts and titles to identify potentially eligible studies. The full text of these potentially eligible studies was then evaluated. Agreement of at least two reviewers was required for decisions regarding inclusion or exclusion of studies. Two authors (E.A. and N.P.) independently assessed the risk of bias and quality of studies in each eligible trial. The full text of these potentially eligible studies was then evaluated. Study quality was formally evaluated using the National Heart, Lung, and Blood Institute Quality Assessment Tool for Case Series Studies (Appendix A) [18]. An agreement between the two reviewers was mandatory for the final classification of studies.

### 2.5. Data Analysis and Synthesis

We used software (StatsDirect version 3.2.10, StatsDirect Ltd., Wirral, UK) to pool estimates of all-cause mortality, major and bleeding rates, using both fixed and random effects models for combining proportions. In the absence of heterogeneity, data were analyzed using the Mantel–Haenszel method. A DerSimonian–Laird random-effects model for pooled estimates of odd risks (ORs) with their 95% confidence intervals (CIs) estimation of all outcomes was used. The Freeman–Tukey variant of the arcsine square root transformation was used to account for the fact that proportions with extreme values (close to 0 or 1) have lower variances. The Cochran Q test of heterogeneity and the I2, I-square of inconsistency were used to assess heterogeneity between studies. Statistically significant heterogeneity was defined as an X2, Chi-square *p*-value less than 0.05 or an I2, I-square greater than 75%. Reported values are two-tailed, and hypothesis-testing results were considered statistically significant at *p* < 0.05. We did not perform statistical testing for publication bias due to the small number of included studies (less than ten). No extramural funding was used to support the work. The authors are solely responsible for the design and conduct of this study: all study analyses, the drafting and editing of the paper and its final contents.

## 3. Results

### 3.1. Study Selection

A total of twelve studies meeting the inclusion criteria were identified [14,15,16,17,19,20,21,22,23,24,25,26]. The selection process is illustrated in Figure 1 (PRISMA) and a total population of 17,921 patients with HFpEF who underwent either a rhythm control, including CA, or a rate control strategy, were included. There was an excellent agreement between investigators on the inclusion of the selected studies. The diagnosis of HFpEF differed slightly across studies. Five studies agreed in patients’ selection based on previous guidelines [27,28]. The rest of the studies were mainly based on symptoms of HF, echocardiographic data and diastolic dysfunction except one which did not explicitly mention a clear definition or process in patients’ selection. The most recent guidelines for HFpEF definition are by Pieske et al. [29].

Four studies used for the analysis were prospective, three of which were single-centre in design and one was multi-centre. Three studies were retrospective multi-centre observational and five were retrospective single-centre observational studies. According to the National Heart, Lung, and Blood Institute Quality Assessment Tool [18], there is a maximum of nine criteria which apply for case series as shown in the Appendix A online, six studies fulfilled eight criteria, while only one study fulfilled five criteria. Of note, three studies [14,15,16] compared rhythm control strategies, including CA versus rate control strategies with AAD (however, Kelly et al. notably comprised a significantly small proportion of HFpEF patients undergoing CA within their rhythm control group [14], and Machino-Ohtsuka et al. grouped AAD together with AAD plus CA in their rhythm control group [15]), while Fukui et al. [17] compared CA alone versus rate control with medication, therefore data were extracted when necessary, if available. These studies were used separately for analysis of rhythm versus rate control outcomes. Moreover, six studies [19,21,22,23,24,25] included subgroups of patients with HFrEF who underwent CA and compared outcomes with HFpEF group, although we focused our results only on the group of HFpEF.

### 3.2. Patient Characteristics

The mean age of the patients was 66.4 ± 9.4 years from the available data, 40/60% males/females with a BMI of 25.4 ± 9.3. Baseline of selected trials, comorbidities, baseline characteristics, symptoms class and medication are presented in Table 1. Hypertension prevalence ranged between 39.5 to 85%, and pre-intervention stroke was present in 18.8% patients on average. Three studies mention the duration of AF prior to intervention which on average was 6.6 years while of note, 45.8% based on the existing data presented with paroxysmal AF. In addition, NT-pro-BNP ranged between 35.4 and 1056 pg/mL, while baseline LA and LV diastolic dimension was on average 43.9 ± 5.3 mm and 48.6 ± 7.7 mm, respectively (Table 2).

### 3.3. Procedural Data

There was wide agreement amongst studies involving CA. Overall, AADs were discontinued prior to the procedure. The patients were effectively anticoagulated and if necessary transesophageal echocardiography was performed to exclude any atrial thrombi. The ablation procedures were performed both under conscious sedation and under general anesthesia. Pulmonary vein antrum isolation (PVAI) was performed with a double-lasso technique under the guidance of a 3D mapping system predominantly CARTO 3, Biosense-Webster, USA but also NavX (St Jude Medical, Inc., Minneapolis, MN, USA), EnSite NavX (Abbott Medical, St. Paul, MN, USA) along with Vecchio et al. [19] who used Ensite Velocity cardiac mapping system (St. Jude Medical Inc.). Patients underwent circumferential and/or segmental PVI with or without deployment of linear lesions. The endpoint of the PVAI was the achievement of bidirectional conduction block between the LA and PVs. Radiofrequency current was delivered point-by-point with an externally irrigated-tip ablation catheter. When an arrhythmogenic superior vena cava (SVC) was identified, an electrical SVC isolation was added. In patients with persistent AF, substrate modification was performed systemically targeting AF termination if AF did not terminate during PVAI as described previously. During the repeat procedure if required, the previous lesion set was evaluated and consolidated. Then, any identified non-PV foci were eliminated. Cryoballoon ablation catheters were used only by Eitel et al. [25] in 11% of HFpEF patients versus 87% who underwent radiofrequency ablation. From the available data, total procedural duration time was 198 ± 68 min (Table 2).

### 3.4. Short-Term Outcomes

AF ablation which was the first procedure for 80.3% of the patients was shown to be quite a safe procedure in HFpEF patients with reasonable overall outcomes. Overall, the pooling of our data shows major vascular complications occurred in 0.4% (95% CI 0.00–0.01) (Figure 2) and major bleeding occurred in 0.5% (95% CI 0.00–0.01) (Figure 3) whereas additional, less severe, various in-hospital or procedural complications were noted in 5.7% of HFpEF patients overall (Table 3). There were two cases with periprocedural stroke reported by Eitel et al. (Table 3) [25].

### 3.5. Long-Term Outcomes

Long-term AF recurrence was noticed in 22.3% of the patients. Of note, pooling of our data showed that long-term sinus rhythm was achieved in 58.0% (95% CI 0.44–0.71) (Figure 4) without the use of AAD. However, this did not seem to affect average NYHA class change nor MAFSI symptom frequency, significantly (*p* = NS for both). Subsequent analysis demonstrated that, in patients with HFpEF undergoing CA for AF, admission for HF occurred in 6.2% (95% CI 0.04–0.09) (Figure 5). Moreover, all-cause mortality was identified in 6.3% (95% CI 0.02–0.13) (Figure 6).

## 4. Discussion

The salient findings of our analysis are as follows: (i) CA appears to be a safe option for patients with AF and HFpEF, (ii) most patients included in the individual studies were mildly symptomatic and relatively young with a mean age of 64 years, (iii) approximately two thirds of patients maintained sinus rhythm in the long term and slightly more than one fifth had recurrence of AF, (iv) CA was associated with lower rates of AF and all-cause mortality compared with non-invasive strategies in HFpEF patients with AF. However, these findings should be interpreted in light of the lack of randomized controlled trials and significant heterogeneity among studies. The lower risk of death or cardiovascular readmissions in patients with AF undergoing rhythm control therapy via CA in comparison to those treated with rate control regimens is best presented by the CABANA trial. In this study, a total of 1307 patients out of 2204 underwent CA. The risk for death or cardiovascular hospitalization in the rhythm control group was 51.7%, which was 6.4% lower compared to the rate control population [31]. Another landmark study on the field, the CASTLE AF trial, presented the supremacy of CA over antiarrhythmic drugs as rhythm control in patients with AF and concomitant HF. Patients treated with CA had significantly lower risk for all-cause death, cardiovascular death, and HF related hospitalizations [32].

Regarding the limitations of this study, although rhythm control and more specifically CA seems to represent the treatment of choice in AF patients, there is limited evidence in current literature for the subpopulation of HFpEF patients. Our systematic research yielded only 13 studies related to AF in HFpEF patients. The differences in the design of these studies impeded the inclusion of all available data in the meta-analysis, and only four studies were included in the HF hospitalization meta-analysis [17,21,24]. Other important limitations include a lack of comparator control group when evaluating the effects of CA in patients with HFpEF, the heterogeneity of criteria, methods, and type of data analysis used which may inadvertently lead to misleading inferences regarding the relationship between variables, in addition to the broad range of sample sizes reported amongst studies. Notably, studies based on small samples sizes typically yield lower statistical power and may lead to erroneous inferences when interpreted in meta-analyses. The study conducted by Kelly et al., although comprising the largest population within the meta-analysis, also conferred significant heterogeneity. This is likely due to the fact that it was a registry trial with fewer exclusion criteria, significantly older patients, and substantially more recruited females [14]. Moreover, only 1% of patients on rhythm control were treated with AF ablation or surgery (in hospital), which is notably a point of study bias. In one study by Machino-Ohtsuka et al., HFpEF patients receiving AAD as well as those receiving AAD plus CA were included within the overall rhythm control group, rendering this data open to misinterpretation [15]. In another study by Machino-Ohtsuka et al., a proportion of patients (*n* = 54) were documented as having undergone multiple ablation procedures in addition to pharmaceutical assistance to help achieve ablation success—this is another notable limitation [20].

AF is a comorbid arrhythmia in almost half of HF patients regardless of LVEF [24]. AF is at least partly induced by HF, while the loss of atrial function itself significantly worsens the prognosis of HF patients, creating a lethal vicious circle [28,33]. In a meta-analysis conducted by Briceño et al., CA as a rhythm control treatment of AF in HFrEF patients reduced mortality rates and improved functional status significantly compared to conventional treatment [34]. Taking into consideration the similarities between HFrEF and HFpEF [28], it would be worthwhile to investigate the effects of rhythm control as the default treatment modality for AF in HFpEF patients via multi-centre randomized control trials.

## 5. Conclusions

This meta-analysis is the first to focus on determining the benefits of a rhythm control strategy for patients with AF and HFpEF, evaluating the role of CA on the maintenance of sinus rhythm, and subsequent outcome measures including hospitalization and morbidity and mortality data. Overall, AF ablation, which was the first procedure for 80.3% of the patients, was shown to be a quite safe procedure in HFpEF patients with reasonable overall outcomes and complication rates, while the vast majority remained in sinus rhythm long-term. Long-term AF recurrence was also noticed in 22.3% of patients. We also demonstrated low rates of hospitalization for HF as well as all-cause mortality in HFpEF patients undergoing CA for AF. In conclusion, these data indicate it may be worthwhile to investigate the effects of rhythm control as the default treatment modality for AF in HFpEF patients via multi-centre randomized control trials.

## Figures and Tables

**Figure 1 jcm-11-00288-f001:**
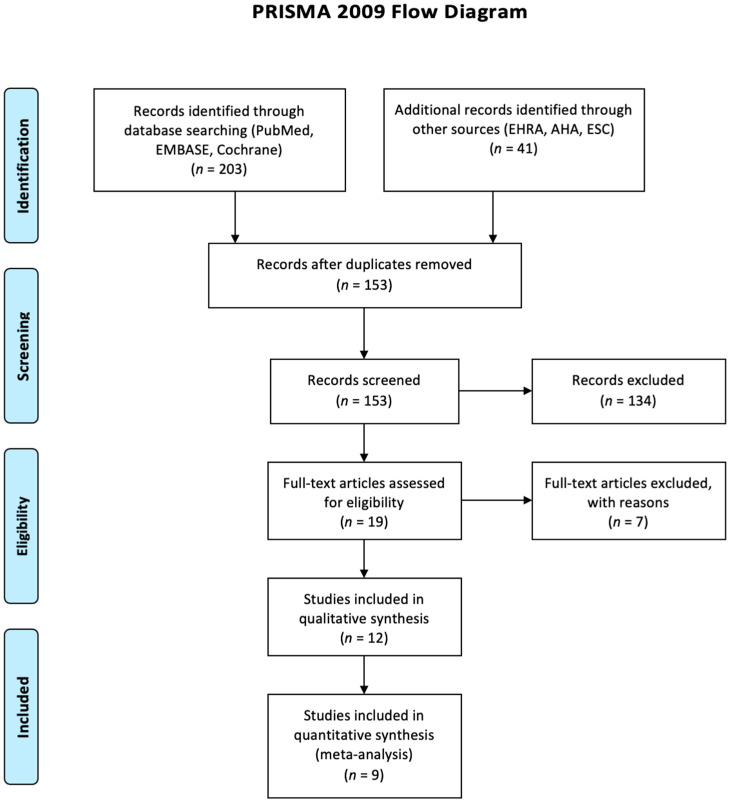
PRISMA flow diagram of study selection.

**Figure 2 jcm-11-00288-f002:**
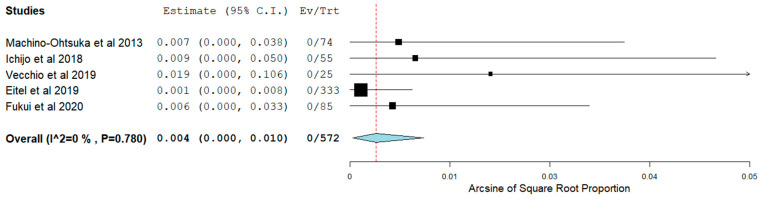
Major vascular complications in patients with HFpEF.

**Figure 3 jcm-11-00288-f003:**
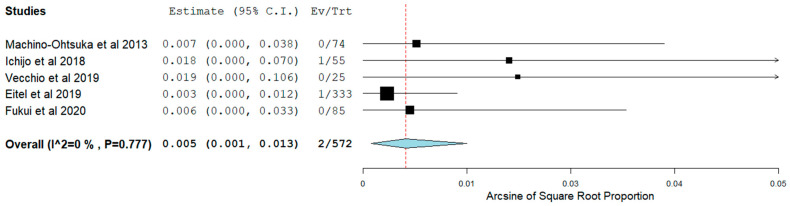
Major bleeding post-ablation in patients with HFpEF.

**Figure 4 jcm-11-00288-f004:**
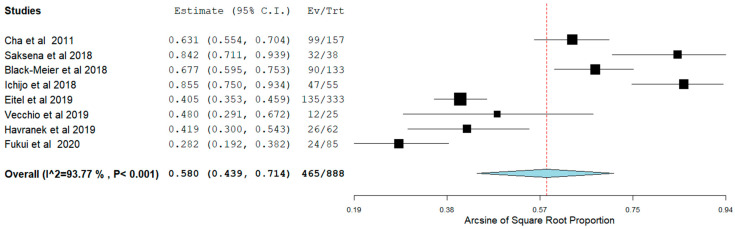
NSR post-ablation without AAD.

**Figure 5 jcm-11-00288-f005:**
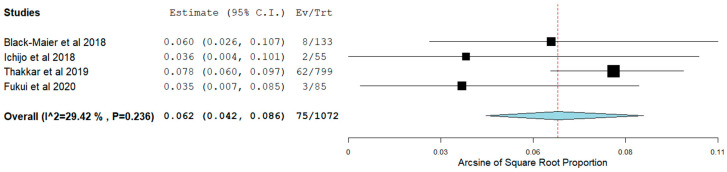
HF admissions in patients with HFpEF.

**Figure 6 jcm-11-00288-f006:**
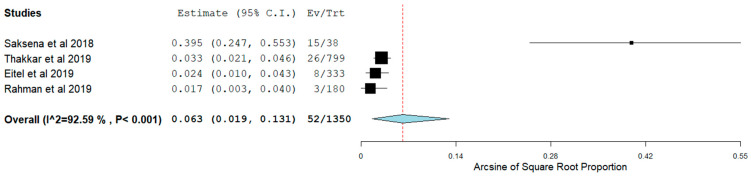
All-cause mortality in patients with HFpEF.

**Table 1 jcm-11-00288-t001:** Baseline characteristics of selected trials.

Authors	Patients (*N*)	Study Design	HFpEF Inclusion Criteria	Age (Mean ± SD)	Female *N* (%)	BMI (Mean ± SD)	HTN, *N* (%)	DM, *N* (%)	IHD, *N* (%)	Stroke, *N* (%)	B-Blockers *N* (%)	CCB, *N* (%)	Digoxin, *N* (%)	AAD, *N* (%)
Cha (2011)	157	Prospective, single-centre	LVEF ≥ 50% and abnormal diastolic function	62.2 (54.4, 70.5)	50 (31.8)	N/A	75 (47.8)	15 (9.6)	27 (17.2)	8 (5.1)	102 (65.0)	31 (19.7)	N/A	85 (54.1)
Machino-Ohtsuka (2013)	74	Prospective, single-centre	LVEF > 50% and fulfilled criteria for HFpEF according to the European Society of Cardiology recommendations [27]	65.0 ± 7.0	19 (25.7)	26.7 ± 14.7	57 (77.0)	21 (28.4)	14 (18.9)	10 (13.5)	53 (71.6)	34 (45.9)	5 (6.8)	Class I = 57 (77.0) Class III = 37 (50.0) Class IV = 12 (16.2)
Black-Maier (2018)	133	Retrospective, single-centre	LVEF ≥ 50%	68.0 (60.0, 74.0)	56 (42.1)	32.0 (28.0, 38.0)	113 (85.0)	38 (28.6)	N/A	N/A	97 (72.9)	N/A	20 (15.0)	Class 1C = 10 (7.5) Class III = 73 (54.9) Amiodarone = 16 (12.0%)
Ichijo (2018)	55	Prospective, single-centre	LVEF > 45% [28]	64.0 ± 10.0	11 (20.0)	25.5 ± 4.7	33 (60.0)	13 (23.6)	10 (18.2)	5 (9.1)	33 (60.0)	15 (27.3)	N/A	24 (43.6)
Kelly (2019)	15,682 (1857 patients in the rhythm control group)	Retrospective, multi-centre	LVEF ≥ 50% or normally/mildly impaired systolic function classified as HFpEF as characterised in the GWTG-HF analyses [30]	81.0 *	1222 (65.8)	N/A	1556 (83.8)	669 (36.0)	904 (48.7)	325 (17.5)	N/A	N/A	N/A	N/A
Machino-Ohtsuka (2019)	158 (79 patients in the rhythm control group)	Retrospective, multi-centre	Fulfilled criteria for HFpEF according to guidelines [24,25]	68.0 ± 7.0	32 (40.5)	24.6 ± 4.2	59 (74.7)	27 (34.1)	13 (16.5)	10 (12.7)	53 (67.1)	34 (43.0)	N/A	Class Ia = 5 (6.3) Class Ic = 31 (39.2) Amiodarone = 44 (55.7) Aprindine = 8 (10.1)
Eitel (2019)	333	Prospective, multi-centre	LVEF ≥ 50% [28]	65.4 ± 9.6	113 (33.9)	N/A	255 (76.7)	36 (10.8)	151 (45.3)	24 (7.1)	240 (72.1)	N/A	N/A	Classes I, III, IV = 177 (53.2)
Fukui (2020)	85 (35 patients in the catheter ablation group)	Retrospective, single-centre	LVEF ≥ 50% with LV diastolic dysfunction	70.0 ± 8.0	12 (34.3)	N/A	21 (55.0)	8 (21)	N/A	N/A	20 (57.0)	N/A	N/A	Amiodarone = 14 (40)

Data presented as median (IQR) or mean ± standard deviation. N/A: data not available; GWTG-HF: get with the guidelines—heart failure. * Median value provided.

**Table 2 jcm-11-00288-t002:** Specific procedural and patient characteristics of selected trials.

Authors	Duration of AF Prior to Intervention (Years ± SD)	AF Type *N* (%)	Pre-LVEF (%, Mean ± SD)	LA Volume (Mean ± SD)	E/E’ (Mean ± SD)	Treated Using Catheter Ablation *N* (%)	First Procedure, *N* (%)	Radiofrequency *N* (%)	Circumferential PVI, *N* (%)	3D Mapping System	Procedure Time (min, Mean ± SD)
Cha (2011)	4.2 (1.7, 8.5)	Paroxysmal = 78 (49.7) Non-paroxysmal = 79 (50.3)	62.0 [60.0, 65.0]	40 cm^3^/m^2^ [35, 50]	12.0 [8.6, 15.7]	157 (100)	138 (88.0)	157 (100)	157 (100) (PVI and WACA)	N/A	94.0 (57.0, 133.0)
Machino-Ohtsuka (2013)	7.3 ± 7.2	Paroxysmal = 23 (31.0) Persistent = 7 (9.5) Long-standing = 44 (59.5)	66.7 ± 7.2	Baseline = 45.2 ± 17.5 mL/m^2^ Follow-up = 42.6 ± 20.2 mL/m^2^	Baseline = 11.8 ± 4.7 Follow-up = 10.3 ± 3.7	74 (100)	24 (32.4)	N/A	N/A	N/A	N/A
Black-Maier (2018)	N/A	Paroxysmal = 45 (37.2) Non-paroxysmal = 76 (62.8)	55.0 (55.0, 55.0)	N/A	N/A	133 (100)	127 (95.5)	133 (100)	133 (100)	CARTO (Biosense-Webster Inc, Diamond Bar, CA) or NavX (St Jude Medical, Inc, Minneapolis, MN)	233.0 (192.0, 290.0)
Ichijo (2018)	N/A	Paroxysmal = 23 (41.8) Non-paroxysmal = 32 (58.2)	57.0 ± 8.0	N/A	N/A	55 (100)	N/A	55 (100)	N/A	CARTO 3 (Biosense-Webster, Irvine, CA, USA)	N/A
Kelly (2019)	N/A	N/A	58.0 *	N/A	N/A	19 (1)	N/A	N/A	N/A	N/A	N/A
Machino-Ohtsuka (2019)	5.0 ± 5.3	Paroxysmal = 34 (43.0) Non-paroxysmal = 45 (57.0)	65.0 ± 8.0	51.0 ± 21.0 mL/m^2^	12.0 ± 4.6	66 (83.5)	N/A	N/A	N/A	N/A	N/A
Eitel (2019)	N/A	Paroxysmal = 153 (45.8) Persistent = 136 (41.0) Permanent = 44 (13.3)	N/A	N/A	N/A	333 (100)	271 (80.2)	294 (87.0)	282 (83.4)	N/A	175.8 ± 77.8
Fukui (2020)	N/A	Paroxysmal = 14 (40) Non-paroxysmal = 21 (60.0)	62.0 ± 8.0	N/A	16.0 ± 7.0	35 (100)	N/A	35 (100)	N/A	CARTO 3 (Biosense Webster, Diamond Bar, CA) or EnSite NavX (Abbott Medical, St. Paul, MN)	168.0 ± 45.0

Data presented as median (IQR) or mean ± standard deviation. N/A: data not available. * Standard deviation not specified.

**Table 3 jcm-11-00288-t003:** Mortality and complication related outcomes of selected trials.

Authors	Follow-Up (Months)	Major Bleeding *N* (%)	Vascular Complications, *N* (%)	Stroke, *N* (%)	Total Complications *N* (%)	AF Recurrence *N* (%)	Patients in SR *N* (%)	Change in Symptoms	HF Admission *N* (%)	All-Cause Admission, *N* (%)	Death/All-Cause Mortality, *N* (%)
Black-Maier (2018)	10.3 (7.3, 12.1)	Peri-procedural = 4 (3.0) (access site bleeding)	Peri-procedural = 0 (0)	Peri-procedural = 0 (0)	Peri-procedural = 9 (6.8)	43 (33.9)	90 (67.7)	MAFSI symptom severity = −0.23 MAFSI symptom frequency = −1.05	8 (6.0)	35 (26.3)	N/A
Ichijo (2018)	32.8 ± 18.5	Post-procedure = 1 (1.8)	Post-procedure = 0 (0)	Post-procedure = 0 (0)	Procedural = 3 (5.5) Post-procedure = 1 (1.8)	8 (14.5)	47 (85.5)	N/A	2 (3.8)	N/A	N/A
Kelly (2019)	12 *	Rhythm = 79 (4.3) Rate = 655 (4.7)	N/A	Rhythm = 29 (1.6) Rate = 318 (2.3)	Rhythm = 680 (36.6) Rate = 4858 (35.1)	N/A	N/A	N/A	Rhythm = 488 (26.3) Rate = 3830 (27.7)	Rhythm = 1151 (62.0) Rate = 8931 (64.6)	Rhythm = 572 (30.8) Rate = 5184 (37.5)
Machino-Ohtsuka (2019)	24 (11–37)	0(0)	0 (0)	0 (0)	0 (0)	Rhythm = 22 (27.8) Rate = 75 (94.9)	Rhythm = 57 (72.2) Rate = 4 (5.1)	N/A	Rhythm = 5 (6.3) Rate = 18 (22.8)	N/A	Rhythm = 2 (2.5) Rate = 8 (10.1)
Eitel (2019)	12	In-hospital = 7 (2.1) Post-procedure = 1 (0.3)	In-hospital = 8 (2.4) Post-procedure = 0 (0)	In-hospital = 2 (0.6) Post-procedure = 4 (1.3)	In-hospital = 41 (12.3) Post-procedure = 7 (2.2)	140 (47.9)	Without AADs = 135 (49.1)	N/A	N/A	150 (50.0)	8 (2.5)
Rahman (2019)	12	N/A	N/A	Rhythm = 2 (2.4) Rate = 9 (9.5)	Rhythm = 2 (2.4) Rate = 9 (9.5)	Rhythm = 16 (18.8) Rate = 63 (66.3)	Rhythm = 69 (81.2) Rate = 32 (33.7)	N/A	N/A	Rhythm = 51 (60.0) Rate = 52 (54.7)	Rhythm = 3 (3.5) Rate = 2 (2.1)
Fukui (2020)	24	0 (0)	0 (0)	0 (0)	0 (0)	Rhythm = 11 (26.0) Rate = N/A	Rhythm = 24 (68.6) Rate = N/A	N/A	Rhythm = 3 (8.6) Rate = 24 (48.0)	N/A	N/A

Data presented as median (IQR) or mean ± standard deviation. N/A: data not available. * Standard deviation not specified.

## Data Availability

The data presented in this study are available on request from the corresponding author.

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
