# Peer review of "Catheter Ablation for Atrial Fibrillation in Patients with Heart Failure with Preserved Ejection Fraction: A Systematic Review and Meta-Analysis"

_jcm, 2022, doi:10.3390/jcm11020288_

Round 1

Reviewer 1 Report

The present review manuscript described the efficacy of catheter ablation as a rhythm control strategy for heart failure with preserved ejection fraction (HFpEF) patients with atrial fibrillation (AF). I agree with the conclusion. However, there are several questions that should be addressed.

Major

  1. The authors should discuss about the limitation and the negative effect of AF ablation. For example, HFpEF patients with long-standing AF might not be treatable.

  1. As pointed out in the manuscript, the study included patients around 65 years old. Is there any evidence in elder HFpEF patients with AF?

  1. Are there any risk factors predicting patients without their mortality improved even after successful AF ablation and sinus rhythm maintained? How about patients with severe functional mitral and tricuspid regurgitation (Circ J. 2018;82:1451-1458.)?

Author Response

Reviewer 1

The present review manuscript described the efficacy of catheter ablation as a rhythm control strategy for heart failure with preserved ejection fraction (HFpEF) patients with atrial fibrillation (AF). I agree with the conclusion. However, there are several questions that should be addressed.

 Major

  1. The authors should discuss about the limitation and the negative effect of AF ablation. For example, HFpEF patients with long-standing AF might not be treatable.

-We appreciate the reviewer's comments, and we agree this is an important point. We have now included the following text to highlight this limitation.

Page 2/Lines 69-72

  1. As pointed out in the manuscript, the study included patients around 65 years old. Is there any evidence in elder HFpEF patients with AF?

-We thank the reviewer for this useful comment. To help address this query, we have reviewed the most recent ESC guidelines (2021) and relevant important literature on the management of heart failure. However, there do not appear to be any new significant studies regarding catheter ablation for AF in HFpEF with a patient cohort age >65 years, which would add an important aspect in this manuscript, therefore we have not made any major change relevant to this point.

  1. Are there any risk factors predicting patients without their mortality improved even after successful AF ablation and sinus rhythm maintained? How about patients with severe functional mitral and tricuspid regurgitation (Circ J. 2018; 82:1451-1458.)

-We appreciate the reviewer's comments. We agree with this important point. We have amended the manuscript accordingly discussing this point having included the above-mentioned study.

Page 2/Lines 62-65

Reviewer 2 Report

The aim of this systematic review and meta-analysis was to investigate the effects of catheter ablation on outcomes of patients with AF and HFpEF, such as functional status, morbidity and mortality, based on data from observational studies.

The ideal of study is important, however, there are some limitations of this study about the study design and statistical methodology. The authors need to clarify these clearly.

1.    First, the main purpose of this study was to see the effects of catheter ablation on outcomes of patients with AF and HFpEF, which should be the focus of this study. However, the studies used in this meta-analysis included the rhythm control, not catheter ablation patients. i.g. the study of Kelly et al (2019) and Machino-Ohtsuka, et al (2013), the rhythm control was mostly based on antiarrhythmic medication, not catheter ablation. The accurate number of patients for catheter ablation shall be clarified in the original study. It is not appropriate to enroll these studies into meta-analysis to evaluate the effect of catheter ablation as rhythm control therapy. This is major bias of this study.

2.    The authors indicated the outcomes after rhythm control therapy included AF recurrence post ablation, or sinus rhythm maintenance with or without AAD. The end points of the study shall be clarified and uniformly analyzed in the meta-analysis. Also, the functional status, admission due to heart failure, and mortality shall be consistent for analysis. It confused the reviewer and readers.

3.    The inclusion criteria for HFpEF and AF rhythm control for each study should be standardized prior to analysis  

Minor issues:
1.      Egg’s test for small study effect is not seem; besides, the study number is too small, and the studies were not well identified.

2.      Figures are hard to interpret: comparative groups shall be marked on the figures. For Figure 2, Figure 3, and Figure 4, it is confusing for what are you comparing? (AF ablation v.s. medical treatment? the effect direction?)

3.      Typos:
I2, I-square: not I2 statistic
X2, chi-square: not X2 statistic

Author Response

Reviewer 2

The aim of this systematic review and meta-analysis was to investigate the effects of catheter ablation on outcomes of patients with AF and HFpEF, such as functional status, morbidity and mortality, based on data from observational studies.

The ideal of study is important, however, there are some limitations of this study about the study design and statistical methodology. The authors need to clarify these clearly.

  1. First, the main purpose of this study was to see the effects of catheter ablation on outcomes of patients with AF and HFpEF, which should be the focus of this study. However, the studies used in this meta-analysis included the rhythm control, not catheter ablation patients. e.g. the study of Kelly et al (2019) and Machino-Ohtsuka, et al (2013), the rhythm control was mostly based on antiarrhythmic medication, not catheter ablation. The accurate number of patients for catheter ablation shall be clarified in the original study. It is not appropriate to enroll these studies into meta-analysis to evaluate the effect of catheter ablation as rhythm control therapy. This is major bias of this study.

-We would like to thank the reviewer for the very crucial points and directions. We agree this point is of paramount importance. For Kelly et al (2019), rhythm control is defined as use of an ‘antiarrhythmic medication, cardioversion, or AF ablation’. Although we acknowledge that the minority of patients on rhythm control were treated with AF ablation, due to the scarcity of available literature, we opted to include additional studies having evaluated outcomes after catheter ablation in addition to pharmacological management as rhythm control. For clarity and to avoid confusion, we have included the following text within the main manuscript to acknowledge this bias:

Page 10/Lines 283-285

-For Machino-Ohtsuka et al. (2019), all patients underwent catheter ablation. However, there was indeed a proportion of patients having undergone additional pharmacological assistance to help achieve ablation success. Therefore, we have included the following text within the main manuscript for clarity:

Page 10/Lines 285-287

  1. The authors indicated the outcomes after rhythm control therapy included AF recurrence post ablation, or sinus rhythm maintenance with or without AAD. The end points of the study shall be clarified and uniformly analyzed in the meta-analysis. Also, the functional status, admission due to heart failure, and mortality shall be consistent for analysis. It confused the reviewer and readers.

-We appreciate the reviewer's comments. We agree with this important point. For clarity, we have now included data regarding long-term AF recurrence within the abstract and conclusion, so that this information is now consistently reported throughout the manuscript (alongside sinus rhythm maintenance). We have removed text regarding function status from section 3.2 (page 5) to hopefully avoid confusion to the reader. Regarding admissions due to heart failure, this information is indeed detailed in Table 3, and the findings for this have been consistently reported throughout the manuscript. Regarding mortality, we have amended the final sentence of section 3.4 (i.e., “Short-term Outcomes”; page 7) regarding mortality for Eitel et al., to help ensure consistency and clarity.

  1. The inclusion criteria for HFpEF and AF rhythm control for each study should be standardized prior to analysis.

-We thank the reviewer for the important comment. For clarity, we have amended the text for section 2.2 from ‘Study Selection’ to ‘Inclusion Criteria’ (i.e., prospective and retrospective cohort studies published as original research articles). We have now also included a separate ‘Exclusions Criteria’ section (section 2.3), which helps to describe the standardised selection criteria and protocol that was used for all studies prior to statistical analysis.

Minor issues:

  1. Egg’s test for small study effect is not seem; besides, the study number is too small, and the studies were not well identified.

-We appreciate the reviewer's comments. In the revised manuscript we amended the Methods section as follows: "We did not perform statistical testing for publication bias due to the small number of included studies (less than ten).”

  1. Figures are hard to interpret: comparative groups shall be marked on the figures. For Figure 2, Figure 3, and Figure 4, it is confusing for what are you comparing? (AF ablation v.s. medical treatment? the effect direction?)

-We appreciate the reviewer's comments, and we agree this is a very important point. Figures 2, 3 and 4 present the results of a proportion meta-analysis. This is not an analysis with two comparable groups. The studies included herein had only one group of patients, namely those who underwent ablation procedures. Therefore, we used a Freeman-Tukey transformation (arcsine square root transformation) to calculate the weighted summary proportion under the fixed and random effects model. We produced forest plots that demonstrate: i) maker size and confidence intervals representative of study weights, ii) pooled effects representing pooled incidence of specific complications or outcomes, iii) summary statistics of heterogeneity among studies. There is no effect direction presented in any of these figures. The purpose is to provide a pooled incidence of events from individual studies. Again, comparison groups were not available in these studies, hence a different approach to depict pooled effects was selected.

  1. Typos:

I2, I-square: not I2 statistic

X2, chi-square: not X2 statistic

We appreciate the reviewer’s comments. This has now been corrected.

Round 2

Reviewer 1 Report

The response and the correction of the manuscript was convincing. There are no additional questions and revisions to be addressed.

Author Response

We have previously replied to these comments

Reviewer 2 Report

I am not satisfied with the response of the author. Again, the purpose of this study was to observe the effects of catheter alation on the outcome in patients with HFpEF. The main studies enrolled were based on rhythm control, including the pharmacologic control and catheter ablation. eg. the study by Kelly et al (2019) and Machino-Ohtsuka, et al (2013). There were two major limitationsp of this study. First, for traditional meta-analysis, the effect based on a specific therapy approach shall be separated independently for comparations. To show a pooled effect may confound the outcome. Second, the study number to perform the meta-analysis is limited. The author should acquire the exact patient number of AF ablation and event number from a specific study, which was necessary for this study.

Minor issues:
1.Due to the small study effect, the Egger’s test shall be used, which is not shown in this version.
2. In the figure, please mark the groups for direction of effect size.

Author Response

Corresponding Author:

Dr Emmanuel Androulakis MD MSc PhD FESC FEACVI

Royal Brompton and Harefield NHS Foundation Trusts

6rd December 2021

We would be grateful if you would reconsider the revised version (R3) of our manuscript entitled “Catheter Ablation for Atrial Fibrillation in Patients with Heart Failure with Preserved Ejection Fraction: A Systematic Review and Meta-Analysis” for consideration and publication in the Journal of Clinical Medicine for the special Issue "State of the Art in Management of Atrial Fibrillation".

In this revision, we have responded to all queries raised by the reviewers and have provided a point-by-point answer for each.

We look forward to receiving your response.

Yours Sincerely,

Dr Emmanuel Androulakis (corresponding author)

Editorial Board Member J Clin Med

Comments:

I am not satisfied with the response of the author. Again, the purpose of this study was to observe the effects of catheter alation on the outcome in patients with HFpEF.

  1. The main studies enrolled were based on rhythm control, including the pharmacologic control and catheter ablation. eg. the study by Kelly et al (2019) and Machino-Ohtsuka, et al (2013).
    We appreciate the reviewers’ comments, and directions. We agree that the study by Kelly et al (2019) was disadvantages by the fact that a very small proportion of patients with HFpEF had undergone AF ablation within the ‘rhythm control’ group. We have highlighted this as a particular weakness within our study and have now excluded this from our analyses to avoid misinterpretation (this manuscript also unfortunately does not provide data regarding the true number of HFpEF patients that had undergone AF ablation and that had also experienced specific outcomes such as HF hospitalisation and mortality).

    Regarding the study by Machino-Ohtsuka et al (2013), the authors enrolled 74 patients with concomitant HFpEF and AF. All patients underwent CA, after which the authors investigated the recurrence of AF over a 12-month period. These patients were then divided into two groups based on AF recurrence (AF post-CA group and SR (sinus rhythm) post-CA group). All AADs were discontinued for five half-lives before the procedure, with the exception for amiodarone, which was discontinued for at least six weeks before. Successful CA was defined as the maintenance of SR without AADs post-procedure during the follow-up period and excluding the blanking period.

    Therefore, presumably the reviewer is referring the study by Machino-Ohtsuka (2019); we acknowledge that the authors grouped both patients receiving ‘AAD alone’ and those receiving ‘AAD together with CA’ into their ‘rhythm control’ group, which is a source of significant bias with respect to subsequent analyses. To avoid misinterpretation, we have therefore excluded this study from our analyses, since the true number of HFpEF patients with AF undergoing CA and having experienced specific outcomes at follow-up are unfortunately not provided from the manuscript/authors.

  1. There were two major limitations of this study. First, for traditional meta-analysis, the effect based on a specific therapy approach shall be separated independently for comparations. To show a pooled effect may confound the outcome.
    We appreciate the reviewers’ comment. We can confirm that we have now isolated the specific intervention (i.e., CA) in only those studies providing this data and have independently analysed this with respect to the specific outcomes that were reported at follow-up. We have excluded studies such as Kelly et al (2019) and Machino-Ohtsuka et al (2019) since they do not provide separate outcomes for HFpEF AF patients within their ‘rhythm control’ group that had specifically undergone CA. We thank the reviewer for highlighting this important point.

  1. Second, the study number to perform the meta-analysis is limited. The author should acquire the exact patient number of AF ablation and event number from a specific study, which was necessary for this study.
    We appreciate the reviewer’s comment. We certainly agree that the total cohort number within this study was limited with respect to HFpEF patients with AF undergoing specifically CA. As requested, we can confirm that we have acquired the exact number of patients undergoing AF ablation for each specific study and have included this within the column ‘Treated using catheter ablation’ of Table 2. We hope this is adequate and helps to clarify the exact number of patients undergoing CA to the reviewer and reader.

Minor issues:

  1. Due to the small study effect, the Egger’s test shall be used, which is not shown in this version.
    We appreciate the reviewer’s very useful comment. For continuous outcomes with intervention effects measured as mean differences, the Egger’s test may indeed be used to test for funnel asymmetry. However, its use with substantially fewer than 10 studies is generally not recommended in line with the Cochrane Handbook for Systematic Reviews of Interventions.

  1. In the figure, please mark the groups for direction of effect size.
    We appreciate the reviewer’s very useful comment. We feel with the update of the new Figures, the effect size direction should now be clearer and cause no misinterpretation.